# Global DNA Methylation and mRNA-miRNA Variations Activated by Heat Shock Boost Early Microspore Embryogenesis in Cabbage (*Brassica oleracea*)

**DOI:** 10.3390/ijms23095147

**Published:** 2022-05-05

**Authors:** Congcong Kong, Henan Su, Siping Deng, Jialei Ji, Yong Wang, Yangyong Zhang, Limei Yang, Zhiyuan Fang, Honghao Lv

**Affiliations:** 1Key Laboratory of Biology and Genetic Improvement of Horticultural Crops, Ministry of Agriculture, Institute of Vegetables and Flowers, Chinese Academy of Agricultural Sciences, Beijing 100081, China; 13121238399@163.com (C.K.); 18810835083@163.com (H.S.); dengsiping@163.com (S.D.); jijialei@caas.cn (J.J.); wangyong03@caas.cn (Y.W.); zhangyangyong@caas.cn (Y.Z.); yanglimei@caas.cn (L.Y.); fangzhiyuan@caas.cn (Z.F.); 2Institute of Horticulture, Henan Academy of Agricultural Sciences, Zhengzhou 450002, China

**Keywords:** microspore culture, DNA methylation, small RNA, differentially expressed genes

## Abstract

Microspore culture, a type of haploid breeding, is extensively used in the cultivation of cruciferous crops such as cabbage. Heat shock (HS) treatment is essential to improve the embryo rate during the culture process; however, its molecular role in boosting early microspore embryogenesis (ME) remains unknown. Here we combined DNA methylation levels, miRNAs, and transcriptome profiles in isolated microspores of cabbage ‘01-88’ under HS (32 °C for 24 h) and normal temperature (25 °C for 24 h) to investigate the regulatory roles of DNA methylation and miRNA in early ME. Global methylation levels were significantly different in the two pre-treatments, and 508 differentially methylated regions (DMRs) were identified; 59.92% of DMRs were correlated with transcripts, and 39.43% of miRNA locus were associated with methylation levels. Significantly, the association analysis revealed that 31 differentially expressed genes (DEGs) were targeted by methylation and miRNA and were mainly involved in the reactive oxygen species (ROS) response and abscisic acid (ABA) signaling, indicating that HS induced DNA methylation, and miRNA might affect ME by influencing ROS and ABA. This study revealed that DNA methylation and miRNA interfered with ME by modulating key genes and pathways, which could broaden our understanding of the molecular regulation of ME induced by HS pre-treatment.

## 1. Introduction

Cabbage (*Brassica oleracea* var. *capitata*), a member of the cruciferous family, is an important vegetable crop. Most commercial cabbage cultivars are F_1_ hybrids to ensure high uniformity and yield [1]. To produce hybrids, high-generation inbred lines are needed, and the parental inbred lines are usually produced through a labor-intensive traditional method-system selection [2,3]. Microspore culture is an effective technique to produce doubled haploid (DH) lines for F_1_ hybrid production, which accelerates the breeding process significantly [4,5].

Microspore embryogenesis (ME) is a complex and interesting process in which isolated microspores are induced to form haploid embryos in vitro, and pre-treatment is essential for embryo production [6]. As the first 8 h of culture at 32 °C was found to be critical in determining the developmental fate of the isolated microspores in *B*. *napus*, most molecular-level research to date has focused on this period [7,8]. Breakthrough progress has also been made in isolated microspore culture of *B*. *oleracea* vegetables in recent years. In cabbage, the ME was significantly enhanced at 32.5 °C for 24 h [9]. In broccoli, the combination of cold pre-treatment (4 °C) for 1 or 2 days and heat shock (32.5 °C) for 1 day signifificantly enhanced microspore embryogenesis efficiency compared with cold pre-treatment alone [10]. These studies suggested that heat shock (HS) plays an important role in *B*. *oleracea*. For HS stress, the activation of heat shock proteins (HSPs) were directly linked to heat-responsive pathways [11]. HSPs were initially thought to have a prominent role in microspore induction, but it was found that uninduced microspores exposed to HS also accumulated HSPs as well [12]. Currently, the view is that HSPs may have a role more directly related to stress tolerance [13]. However, microspore culture of cabbage mainly focused on the optimization of culture conditions, while few studies have been carried out on the mechanism of heat shock-induced ME.

In plants, epigenetic modifications play a crucial role in responding quickly to environmental stress. DNA methylation, representing an epigenetic mechanism, regulates gene expression, which is significant for seed/embryo development, gametophyte development, and stress responses [14,15,16,17]. DNA methylation can be categorized into three contexts: symmetric CG and CHG (where H = A, T, or C) and asymmetric CHH [18,19]. Methylation modification is stable and heritable, but it may change in response to development and the environment [20]. Recently, it has been found that cellular reprogramming in ME involved in DNA methylation levels decrease with the activation of cell proliferation, and subsequently increases with embryonic differentiation in some plant species, like *Brassica napus* [21], *Hordeum vulgare* [22], and *Quercus suber* [23]. A recent study also demonstrated that disrupted genome methylation in response to high temperature (39−41 °C in the daytime and 29−31 °C at night) has distinct effects on microspore abortion and anther indehiscence in cotton, and transcriptome analysis indicated that high temperature disturbs sugar and ROS metabolism via disrupting DNA methylation [24]. For *B. napus*, studies have shown treatment on microspores at 32 °C for 6 h triggered DNA hypomethylation, and the total number of differentially methylated region-related genes at 32 °C for 6 h was about twice as high as that at 18 °C for 6 h [25]. Heat pre-treatment is one of the most important stresses for ME induction [13], apart from this, culture of isolated microspores of *B*. *oleracea* at 25 °C has been proposed as an optimal system to study the gametophytic development in vitro [9].

Additionally, microRNA (miRNA), a class of small non-coding RNAs approximately 18–25 nt in length, is another important epigenetic regulator that post-transcriptionally represses gene expression in a wide variety of eukaryotic organisms [26,27]. As a major type of small RNAs, miRNA participates in the regulation of diverse plant developmental processes such as seed germination [28], embryogenesis [29], leaf growth [30], and fruit development and ripening [31]. In other reports, the negative regulation of miRNAs through target mRNA degradation or translational repression is crucial for responding to biotic and abiotic stresses [32,33]. To investigate the role of miRNAs in barley ME development, Bélanger et al. [34] detected that the abundance of 51 of these miRNAs differed significantly during microspore development under thermal, osmotic, and starvation stresses, and degradome analysis revealed that the transition of microspore to embryogenesis pathway involved miRNA-directed regulation of members of the ARF, SPL, GRF, and HD-ZIPIII transcription factor families. Through small RNA sequencing of buds in cotton, Chen et al. [35] found that expression abundance of miR156 under high temperature stress was 10% higher than the total expression abundance under normal temperature at microspore release stage, which was the main regulators responding to HT stress with positive or negative regulation patterns. Although implicated in an array of embryonic developmental processes, miRNA functions during ME in cabbage remain unclear. Thus, identifying miRNAs and their targets and elucidating their regulatory mechanisms are critical to understand ME in cabbage.

In summary, DNA methylation and miRNA regulation have been shown play an important role in ME development in some plant species, but the epigenetic modification in *B*. *oleracea* microspore toward embryoids is not clear. In this study, we have performed the first integrated genome-wide analysis of DNA methylation, miRNAs, and mRNA transcriptional activity, using HS stress (32 °C for 24 h) and normal temperature stress (25 °C for 24 h) microspores from cabbage accession ‘01-88’. We constructed two DNA libraries, six small RNA libraries, and six mRNA libraries, and the objective were to analyze the landscape of methylome distribution, differentially expressed miRNAs (DERs) and differentially expressed genes (DEGs) involved in cabbage ME development. The work performed in this study will broadens our understanding of the molecular regulation of ME induced by HS stress.

## 2. Results

### 2.1. Microspore Collection and Culture

The isolated microspores from the highly embryogenic inbred line ‘01-88’ were subjected to HS (32 °C) and non-HS (25 °C) pre-treatment for 24 h, respectively. Then after 3 weeks of culture at 25 °C, embryoids were produced only from isolated microspores under 32 °C HS for 24 h (Figure 1A). The average number of embryoids per bud was 47 after 32 °C HS treatment, compared with zero embryoids per bud under 25 °C normal temperature treatment (Figure 1C), and the difference in the embryoids rates between the 32 °C and 25 °C treatments was significant (*p* < 0.01).

### 2.2. The DNA Methylation Landscapes of Microspores Treated at Two Temperatures

Genome wide profiling of DNA methylation using bisulphite sequencing (BS-Seq) was performed in microspores treated with heat temperature at 32 °C for 24 h (HT32) and normal temperature at 25 °C for 24 h (NT25), respectively. The two libraries generated a total of 104,763,044 (HT32) and 133,801,556 (NT25) raw reads. Following filtering, 54,750,572 clean reads (52.26%) in HT32 and 69,437,722 clean reads (51.90%) in NT25 were successfully and uniquely aligned to the ‘02-12’ cabbage reference genome (BRAD, http://brassicadb.cn, 12 June 2021). These data were used to retrieve the methylation level for each CG, CHG, and CHH site. A total of 32,351,452 cytosine methylation levels (mCs) (60.8% at CG sites, 31.0% at CHG sites, and 8.2% at CHH sites) and 42,291,691 mCs (60.4% at CG sites, 30.9% at CHG sites, and 8.7% at CHH sites) were identified in HT32 and NT25, respectively (Figure 2). Fisher’s test were used to calculate the significance of the CG, CHG, and CHH methylation numbers and the total methylation numbers between the two pre-treatments, respectively, and the *p* values were all less than 0.01. The total mCs were slightly different between the HT32 (21.15%) and NT25 (21.60%) in the whole cytosine sites; The overall genomic methylation degree of the mCGs was higher in the HT32 samples (69.31%) than in the NT25 samples (68.95%); For the mCHGs, the methylation level showed identical between the HT32 (35.32%) and NT25 (35.32%), whereas for the mCHH, the methylation level was lower for HT32 (9.41%) than for NT25 (9.97%) (Appendix A).

To show the genome-wide DNA methylation landscape of the two samples, DNA methylation maps in the three contexts were represented using a Circos histogram (Figure 3). The outermost circle is a scale presentation based on the corresponding chromosome length, the outer and inner three circles are the background of methylation of CG, CHG, and CHH in the chromosome intervals of HT32 and NT25, and the darker the color, the higher the background level of methylation. We observed some different methylation peaks, and the methylation levels of mCHH in HT32 and NT25 could be clearly distinguished in some local positions of chromosomes, e.g., 3–5 Mb, 14–16 Mb and 25–30 Mb on chromosome 1. Further, we calculated the percentage of DNA methylation levels in each context throughout the nine chromosomes (Appendix A); these nine chromosomes showed similar methylation patterns to the genome.

### 2.3. DNA Methylation Levels in Different Genomic Regions

We analyzed the differential methylation levels of gene bodies (first, internal, and last exons and introns) and their 2 kb up- and downstream regions (distal, intermediate, and proximal promoter; downstream) between HT32 and NT25 (Figure 4). In general, the methylation levels of CG, CHG, and CHH were extremely low at the transcriptional start site (TSS) and transcriptional end site (TES), and the CG, CHG, and CHH methylation levels were higher in the flanking regions than in the gene bodies (Figure 4). Moreover, the CG, CHG, and CHH methylation levels were slightly higher in HT32 than in NT25, except CG in the flanking area, the number of mCs in HT32 and NT25 was similar. Notably, the frequency of mCs in the CG context was higher than that in the CHH and CHG contexts in the region of analysis. These distinct methylation levels revealed that microspores induction under 32 °C treatment could have higher CG, CHG, and CHH methylation in the distal promoter region.

### 2.4. Identification of Differentially Methylated Genomic Regions (DMRs) and DMR-Associated Genes

To investigate specific DNA methylation in cabbage ME, the genomic regions associated with hypermethylation or hypomethylation were profiled. In total, 1038 differentially methylated genomic regions (DMRs) (*p* < 0.05) were detected in the two treatments. The number of hypermethylated and hypomethylated regions was 414 and 624, respectively, all of which were mapped to the chromosomes (Appendix A). As DNA methylation was associated with gene expression, DMRs in the promoter, introns, exons, and downstream regions were defined as DMR-associated genes. In total, 508 DMR-associated genes were identified, which included 212 hypermethylated genes and 296 hypomethylated genes in HT32. GO enrichment analysis was conducted on these DMR-associated genes. We found 15, 10, and 25 terms in cellular components (CC), molecular functions (MF), and biological processes (BP), respectively (Figure 5A). The GO terms of the integral component of membranes in CC, carbohydrate metabolic in BP, and protein binding in MF were most enriched terms. Moreover, the DMR−associated genes were enriched in the KEGG pathways database to identify pathways that were responsive to early microspore embryogenesis. In total, 25 pathways were detected in our study, which included five branches: organismal systems, metabolism, genetic information processing, environmental information processing, and cellular processes. The most abundant pathway in each branch was the endocrine system, amino acid metabolism, translation, signal transduction, and transport and catabolism, respectively (Figure 5B).

### 2.5. Association Analysis with the DNA Methylome and mRNA Profiles

To explore the influence of DNA methylation on the expression of genes, we analyzed the relationship between DMRs and differentially expressed genes (DEGs, |log2FC| >1 and *p* < 0.05) on a genome-wide scale. In transcriptome data, a total of 7226 DEG (4312 upregulated genes and 2914 downregulated genes) transcripts were identified in HT32 and NT25 (Appendix A, Appendix A). GO enrichment analysis was conducted on the DEGs. We found that 52, 99, and 136 functional groups in CC, MF and BP, respectively (Appendix A). Furthermore, the GO terms of the integral component of membrane in CC, the translation in BP, and the structural constituent of ribosome in MF were most enriched terms. These DEGs were involved in certain embryogenesis-related processes, including the response to abscisic acid (ABA), embryo development ending in seed dormancy, ABA-activated signaling pathway, and response to cytokinin (CTK). Furthermore, 2761 DEGs were associated with the corresponding KEGG pathway, and they were primarily enriched in ribosome, plant-pathogen interaction, flavonoid biosynthesis, stilbenoid, diarylheptanoid and gingerol biosynthesis, ribosome biogenesis in eukaryotes, and galactose metabolism. Among them, 329 and 281 DEGs were enriched in ribosome, plant-pathogen interaction (Appendix A).

The general observation was that the hypermethylated DMR-associated genes exhibited lower levels of transcript abundance, and hypomethylated DMR-associated genes exhibited higher levels of transcript abundance relative to all genes. In HT32 and NT25, 59.92% of all DMRs were associated with transcripts, which included 622 transcripts related to the DMRs. Box-plots showing differential expression levels of genes associated with DMRs are displayed (Figure 6A). For hyper-DMRs between HT32 and NT25, 54 transcripts were downregulated, and 64 transcripts were upregulated (Figure 6B). Some of the DMR-associated genes were negatively correlated with the transcript abundance. For the hypo-DMRs, 53 transcripts were downregulated, and 71 transcripts were upregulated (Figure 6B). The results suggested that these DMRs played a role in the regulation of gene expression and might contribute to the differential temperature stress responses.

### 2.6. Validation of Gene Expression

We validated the expression levels of nine genes from the genes that were negatively regulated by microRNA and methylation in HT32 and NT25 using quantitative real-time PCR (qRT-PCR). We found that Bol019334, Bol018627, Bol044765, and Bol037384 were upregulated, while Bol007881, Bol010698, Bol035641, Bol016979, and Bol042216 were downregulated in HT32 (Figure 7). All genes were significant except gene Bol044765 (*p* < 0.01). Overall, the qRT-PCR results were in concordance with results obtained from RNA-seq data analyses, indicating the high quality of our data. The primers used for qRT-PCR are listed in Appendix A.

### 2.7. Association Analysis with the DNA Methylome and miRNA Profiles

To examine the relationship between the extent of DNA methylation and miRNA abundance, we performed small RNA−seq analysis. Six independent small RNA libraries (HT32-1, -2, -3 and NT25-1, -2, -3) were constructed to sequence the small RNAs and generated a total of about 12.17–18.43 million raw reads (Appendix A). In total, 870 miRNAs (including the putative miRNAs) were found in cabbage, and the predicted targets of 63 differentially expressed miRNAs (DERs) were analyzed (Appendix A). Among the 63 differentially expressed miRNAs, 31 miRNAs were up-regulated, and the remaining 32 miRNAs were downregulated in HT32. For 31 up-regulated miRNAs, the number of normalized reads of aly-miR156h-3p_L+1R+2_1ss11CA was very high (5.73-fold), followed by PC-5p-495929_3, bra-miR156b-p3. Most miRNA members of the same family had similar expression profiles. For example, eight miR2592 and four miR156 family members were significantly up-regulated. Three miR394 family members were found to be significantly down-regulated. In addition, 475 predicted targets were found (Appendix A). GO enrichment analysis was conducted on the predicted targets, and they were primarily enriched in meristem initiation, cell differentiation, seed development, and primary shoot apical meristem specification (Appendix A). According to the KEGG pathway analysis (Appendix A), miRNA targets were primarily enriched in plant hormone signal transduction. Then, we identified the interaction network of miRNA and target genes involved in the plant hormone signal transduction pathway (Figure 8). We found that one miRNA could target several genes, and most miRNA members of the same family targeted the same genes, e.g., bolmiR172a targeted 10 genes, and ath-miR394a, bna-miR394a, and aly-miR394a-59 targeted Bol024053.

Next, we investigated the presence of mCs in the miRNA loci. In total, 39.43% of miRNA loci were associated with mCs. Interestingly, there was no CHG locus in miRNAs associated with mCs. The proportion of CG was slightly higher as compared to the CHH-context mCs in the two treatments (Figure 9A). For the mCGs, the methylation level was higher for HT32 than for NT25. In addition, we estimated the abundance of miRNA in differential DMR-associated genes and found that the abundance of miRNAs was significantly enriched in hypo-DMRs (Figure 9B). These results suggest that miRNAs participate in alteration of DNA methylation levels in different treatments.

### 2.8. Association Analysis of the DNA Methylation, miRNA and mRNA in Cabbage Microspore Embryogenesis

To investigate the effects of DNA methylation and miRNA on gene expression, we analyzed DEGs targeted by both DMR and DER in HT32 and NT25. In total, 31 DEGs were targeted by both methylation and miRNA (Appendix A). These target genes encoded proteins involved in diverse cellular processes, including phenylpropanoid biosynthesis, carotenoid biosynthesis, plant hormone signal transduction, and more (Appendix A). Many studies have found that DNA methylation and miRNA negatively regulated gene expression [36]. Although 19 of the 31 genes were targeted by both DMR and DER, but their gene expression has not completely negative correlation with DNA methylation and miRNA regulation, which were relatively complex, and these DEGs might not be completely regulated by DNA methylation and miRNA or only one of them. Therefore, we selected the remaining 12 key genes whose expression levels were negatively correlated with DNA methylation and miRNA regulation (Figure 10). In hypomethylated regions, miRNAs including bna-miR6028, bol-miR172a, and ptc-miR6478 in cabbage were downregulated under 32 °C stress, and the target genes Bol024133 (TIR-NBS-LRR family), Bol044765 (exostosin family), Bol018627 (ethylene-responsive transcription factor 1B-like, AP2), and Bol019334 (peroxidase CB) were upregulated. In hypermethylated regions, miRNAs including ath-miR164, bra-miR164, htu-miR530, and mtr-miR2592 in cabbage were upregulated under 32 °C stress, and the target genes Bol035641 (kinesin motor protein-related), Bol010698 (SNF1 kinase homolog 10), Bol016979 (unknown), Bol042216 (DNA-binding), Bol017014 (atypical CYS HIS rich thioredoxin 5), Bol021292 (glutamate-ammonia ligase), and Bol007881 (plant vesicle-associated membrane) were downregulated. These genes were associated with the ROS response, hormone signal transduction, and defense response, indicating that the ME mechanism during heat shock induction may be similar to the plant–pathogen interaction.

## 3. Discussion

### 3.1. Global Analysis of HS Boosting ME in B. oleracea

In recent years, a large number of studies have suggested that DNA methylation plays a crucial role in the regulation of seed development, fertility, embryogenesis, and stress adaption [37,38,39,40]. The epigenetic marks, such as DNA methylation, may be attributed to the phenotypic consequences such as embryogenesis from isolated microspores. Therefore, it is important to investigate the DNA methylomes of isolated microspore under 32 °C treatment for 24 h to understand the epigenetic regulation of microspore embryogenesis. In this study, the global DNA methylation pattern was profiled in HT32 and NT25. A comparison of HT32 and NT25 indicated that the genomic cytosine methylation levels showed few differences, including a slightly lower methylation level at mCHH and a higher level at mCG in HT32. Hypomethylation is considered a common feature associated with adaptive responses to various stresses [41,42,43]. In this study, there were 124 hypomethylated DMR-associated DEGs in HT32 and NT25, and these genes could be critical in ME of *B*. *oleracea*.

Recently, high throughput sequencing has been widely applied to identify genome−wide miRNAs involved in plant development and environmental stress responses [44,45]. However, the comprehensive identification of miRNAs and their targets involved in cabbage microspore embryogenesis has not been conducted. In this study, 63 differentially expressed miRNAs between isolated microspores subjected to 32 °C (24 h) and 25 °C (24 h) were identified, and 18 differentially expressed miRNAs with more than a three-fold relative change between isolated microspores subjected to 32 °C (24 h) and 25 °C (24 h) were identified. Previous studies demonstrated that diverse miRNAs exist in plants and they might have potential regulatory roles in somatic embryogenesis and seed development. In maize, miR156 regulates embryogenic callus differentiation [46], and it is also responsible for cotyledon embryo development in larch [47]. Overexpression of csi-miR156a significantly enhanced the initial phases of SE induction in preserved citrus embryogenic callus [48]. Both miR156 and miR172 are master regulators of phase transition and seed germination in plants [49]. Auxin homeostasis is important for embryo development and is mediated by the action of miR165/166, miR164, and miR160 [50]. In our study, the expression level of bra-miR156b and aly-miR156h was 4.15 and 5.73, and bra-miR164e was only identified at 32 °C after 24 h. We speculated that these miRNAs might be involved in the microspore embryogenesis process. miRNAs and target genes jointly regulate plant growth and development. During somatic embryogenesis, miRNA genes function in various cellular, physiological, and developmental processes to mediate stress adaptation, developmental regulation, and hormone responses [51]. In this study, miRNA target genes were primarily enriched in plant hormone signal transduction, which is consistent with previous research results [50,51].

A correlation between miRNA abundance and DNA methylation has been established in some studies [52,53]. However, whether miRNA and DNA methylation regulate gene expression in microspore embryogenesis under 32 °C heat-shock stress in cabbage is unknown. In this study, we identified 17 miRNAs and 31 target genes that were differentially methylated under 32 °C HS stress. Several pairs of genes that were negatively regulated by miRNA and methylation may play an important role in ME. Six target genes were involved in the oxidation-reduction process; the reactive oxygen intermediates produced as signaling molecules by these gene products, along with each gene’s role in oxidation-reduction reactions, could control cabbage response to various processes including programmed cell death, abiotic stress response, and systemic signaling [54].

In the induction process of ME in some plants, HSPs could be related to the acquisition of embryogenic potential by microspores, but some scholars hold different views in *B*. *napus*. Seguí-Simarro et al. [12] found that uninduced microspores exposed to HS also accumulated HSPs. Zhao et al. [55] found that there was no specific expression of HSP during microspore culture in colchicin-treated *B*. *napus*, suggesting that HSP may not be necessary for ME. In our mRNA data, there was no significant difference in HSP expression between the two samples, suggesting that HSPs could not be necessary in *B*. *oleracea*, as it was in *B*. *napus*. In miRNA data, bol-miR172a was downregulated to target AP2. In *Arabidopsis*, miR172 associates with AGO1 or AGO10 and acts to silence specific target gene expression; miR172 targets the transcription factor gene AP2, which regulates juvenile to adult-phase transition, floral organ identity, and flowering time [56,57,58]. Interestingly, the expression level of AP2 was 14.95, and the DNA methylation level was hypomethylated in HT32 and NT25. We identified those genes with altered expression levels through transcriptional silencing mechanisms including DNA methylation and miRNA cleavage, and these changes may be correlated with the embryogenesis induced by heat shock treatment.

### 3.2. Heat Shock-Induced DNA Methylation and miRNA Associated–Genes Invovled in the ROS Response and ABA Signal Transduction

According to previous studies [59,60], ME was related to ROS response. Among the 31 DEGs targeted by methylation and miRNA, seven genes could be found to be closely related to early ME (Table 1). According to the GO and KEGG annotation information (Appendix A), two genes (Bol037384 and Bol01933, peroxidases) were involved in the response to oxidative stress and could remove ROS. The two genes were negatively regulated by the same miRNA targeting and methylation in HT32, but were upregulated in the transcriptome. We hypothesized that HS mediated ROS metabolism disrupted DNA methylation and miRNA, leading to protective function in ME.

There are other reports that ME is strongly affected by ABA regulation [61,62]. In our study, Bol007667 (plant U-box 9, PUB9) was involved in the ABA-activated signaling pathway, and Bol016003 (glycine-rich protein 2B) was involved in the ABA response. Reynolds et al. [61] showed that the mechanism of ABA in ME was related to the *EcMt* gene (early cysteine-labeled metallothionein gene). During this process, calcium is involved in ABA signal transduction to initiate the EcMt gene; thus, ME could have a possible relationship with calmodulin [62]. In HT32 and NT25, one atypical CYS HIS sulfur-rich thioredoxin gene (Bol017014) and two calcium-related genes (Bol012681 and Bol045794) were identified, which were regulated by methylation and miRNA.

It is generally believed that methylation and miRNA located at promoters and genes had a greater influence on gene expression. However, the regulation of methylation and miRNA on genes is very complex, which occurs in different regions of genes and may have different functions [63,64]. Therefore, the expression of these genes could be affected by methylation or miRNA and played a regulatory role in ME, which was worthy of further study.

Overall, it would be interesting to study the changes of methylation levels in cabbage microspore under HS. These results could broaden our understanding of the molecular regulation of ME induced by HS pre-treatment, and have a significance for producing DH lines from F_1_ hybrids, which accelerates the breeding process significantly.

## 4. Conclusions

To study the regulatory roles of DNA methylation and miRNA in early ME, we combined DNA methylation levels, miRNAs, and transcriptome profiles in isolated microspores of cabbage ‘01-88’ under HS and normal temperature. For global DNA methylation, 508 DMRs were identified. The association analysis found 59.92% of DMRs were correlated with transcripts and 39.43% of miRNA locus were associated with mCs. Furthermore, a total of 17 miRNAs and 31 target DEGs were found to be differentially methylated. These DEGs were mainly involved in ROS response and ABA signaling, and provided evidence that epigenetic regulation play a crucial role in ME activated by HS.

## 5. Materials and Methods

### 5.1. Plant Materials, Treatments, and Sample Collection

The microspore culture of cabbage accession ‘01-88’ (excellent in the embryo yield under microspore culture) was carried out at the Institute of Vegetables and Flowers, Chinese Academy of Agricultural Sciences, Beijing, China. The microspore isolation and culture procedures were performed following Lv [65] and were slightly modified. Briefly, buds with a length of 3–3.5 mm and an anther-to-petal length ratio of about 3:2 were selected. These buds surfaces were sterilized by 75% ethanol and 8% sodium hypochlorite, then transferred to test tubes with round bottom containing a small amount of B5 medium (3.21 g B5 and 130 g sucrose in 1 L dH_2_O, pH 5.9) and grounded with a glass rod. The isolated microspores were filtered, centrifuged, and resuspended in the NLN-13 medium (1.77 g NLN and 130 g sucrose in 1 L dH_2_O, pH 5.9). Subsequently, a 2 mL microspore suspension adjusted to 1 × 10^5^ cfu/mL was added to each petri dish (60 × 15 mm), and were incubated at heat temperature at 32 °C for 24 h (HT32) and normal temperature at 25 °C for 24 h (NT25), respectively.

A biological replicate required 80 buds, and 26 petri dishes of microspore suspension could be obtained, 20 of which were centrifuged to collect microspores for sequencing, and 6 petri dishes were incubated at 25 °C in the dark for 3 weeks until embryoids were produced. *T* test was used to evaluate the number of embryoids produced under the two treatments, with three replicates per treatment, and the significance difference was *p* < 0.05. Small RNA sequencing and RNA sequencing were performed with three biological replicates per treatment, and both were harvested in the same sample. Bisulfite sequencing processed genomic DNA isolated from HT32 and NT25 (pooled in equal quantity from the two independent biological replicates).

### 5.2. Genomic Methyl Cytosine Library Construction and Bisulphite Sequencing (BS-Seq)

Total DNA was extracted using a Hi-Fast Plant Genomic DNA Kit (GeneBetter Biotech, Beijing, China) following the manufacturer’s procedure. The quality of DNA was evaluated by agarose gel electrophoresis and a spectrophotometer (Thermo Fisher Scientific Inc., Wilmington, DE, USA). The genomic DNA samples were fragmented by sonication to a size of 100–300 bp. The DNA fragments were end repaired and the 3′-end added a single ‘A’ nucleotide. Subsequently, the Accel-NGS Methyl-Seq DNA Library Kit (Vazyme Biotech Co, Nanjing, China) was utilized for attaching adapters to single−stranded DNA fragments. The DNA fragments were subjected to bisulfite conversion using a ZYMO EZ DNA Methylation-Gold kit (NEB, Ipswich, MA, USA). After desalting, size selection and PCR amplification, the two libraries were sequenced by the pair-end 2 × 150 bp sequencing on an Illumina Hiseq 4000 platform (LC Bio, Hangzhou, China).

### 5.3. Bioinformatic Analysis

In-house cutadapt and Perl scripts were used to remove reads with adapter contamination, low quality bases, and undetermined bases [66]. Then, sequence quality was verified using FastQC (http://www.bioinformatics.babraham.ac.uk/projects/fastqc/, accessed on 12 June 2021). Reads that passed quality control were mapped to the reference genome using WALT [67]. After alignment, the reads were further deduplicated using samtool [68]. For each cytosine site (or guanine corresponding to a cytosine on the opposite strand) in the *B*. *oleracea* genome sequence (http://brassicadb.cn, 12 June 2021). The DNA methylation level was determined by the ratio of the number of reads supporting C (methylated) to that of the total reads (methylated and unmethylated) using in-house Perl scripts and MethPipe [69]. Genomic methylation levels of HT32 and NT25 was visualized by circos software [70]. DMRs were calculated using the R package-MethylKit with default parameters (1000 bp slide windows, 500 bp overlap, *p* < 0.05). Multiple testing correction (FDR < 0.05) was followed to test each window. According to the read coverage of methylated C in each sample ≥30, the level ratio of average methylation was >15, the fold change was >2 or <0.5, and the DMR with gene annotation was retained to filter and screen the DMRs [71].

### 5.4. Small RNA Library Construction, Sequencing Analysis, and Prediction of miRNA Target Genes

Total RNAs were isolated separately from the microspores in HT32 and NT25 using Trizol reagent (Invitrogen, GeneBetter Biotech, Beijing, China) following the manufacturer’s protocol. Each treatment had 3 replicates, which were HT32-1, -2, -3 and NT25-1, -2, -3. To construct the six small RNA libraries (HT32-1, -2, -3 and NT25-1, -2, -3), small RNA (18 to 30 nt in length) was fractionated by polyacrylamide gel electrophoresis and ligated to the 5′ and 3′ RNA adapters. Reverse transcription and PCRs were performed to obtain sufficient single-stranded cDNA. Finally, these six small RNA libraries were sequenced by Illumina Hiseq2500 platform (LC Bio, Hangzhou, China).

Raw reads were analyzed according to procedures described in a previous study [72] by ACGT101-miR v4.2 software package (LC Sciences, Houston, TX, USA) to remove adapter dimers and low-quality reads (the reads with more than one base with a Q-value lower than 20). The clean small RNA reads were matched to the GenBank database (https://www.ncbi.nlm.nih.gov/genbank/, accessed on 3 July 2021), Rfam database (https://rfam.xfam.org/, 3 July 2021) and Repbase database (https://www.girinst.org/repbase/) to identify and remove rRNA, scRNA, snRNA and tRNA by blastn (E-value < 10^−5^).

Subsequently, unique sequences with a length in 18–25 nt were mapped to specific species precursors in miRBase 21.0 (ftp://mirbase.org/pub/mirbase/CURRENT/, accessed on 10 July 2021) by a BLAST search to identify known miRNAs and novel 3p- and 5p-derived miRNAs. Length variation at the 3′ and 5′ ends and one mismatch inside of the sequence were allowed in the alignment. The annotated miRNAs were determined as known miRNAs. The unmapped sequences were BLASTed against the specific genomes, and the hairpin RNA structures containing sequences were predicted from the flank 120-nt sequences using RNAfold software (http://rna.tbi.univie.ac. at/cgi-bin/RNAfold.cgi, accessed on 13 July 2021) [73].

Differentially expressed miRNAs (DERs) was analysed based on normalized deep-sequencing counts by *T* test and the significance difference was the FDR adjusted *p* < 0.05. The fold change and *p* value for each miRNA was calculated based on the normalized expression. To predict the genes targeted by DERs, the putative target sites of miRNA were identified using the psRNA Target program (http://plantgrn.noble.org/psRNATarget/, accessed on 12 June 2021) with default parameters, the predicted target genes were evaluated based on complementarity scoring and maximum expectation [74]. The *B*. *oleracea* genome database (http://brassicadb.cn, accessed on 12 June 2021) was used as the sequence library for target searches. The gene functions targeted by miRNAs were clarified using GO (http://www.geneontology.org/, accessed on 15 July 2021) and KEGG (http://www.kegg.jp/kegg, accessed on 12 June 2021) annotation. Significantly enriched GO terms and KEGG pathways were identified using hypergeometric tests and a bonferroni correction with *p* < 0.05 as a threshold [73]. Additionally, the miRNA-pathway interaction network was constructed using Cytoscape v2.8.3 software (http://www.cytoscape.org/, accessed on 15 July 2021).

### 5.5. mRNA Sequencing, Assembly, and Annotation

The quantity of total RNA was determined using a NanoDrop 2000 (Thermo Fisher Scientific Inc., Wilmington, DE, USA), and the integrity was assessed by an Agilent 2100 (Agilent Technologies Co., Ltd., Beijing, China) with RIN number >7.0. Six cDNA libraries (HT32-1, -2, -3 and NT25-1, -2, -3) were prepared and sequencing was performed using the Illumina HiSeq 4000 platform (LC Bio, Hangzhou, China). The low quality reads containing sequencing adaptors, sequencing primers, and/or nucleotides with a quality score (Q < 20) were removed. Then, clean reads were mapped to the reference genome (http://brassicadb.cn, 12 June 2021) by HISAT [75]. Mapped reads were assembled using String Tie [76], and their expression levels were calculated by the FPKM method. Genes differential expression analysis was performed by edgeR between two samples. The genes with the parameter of the FDR adjusted *p* < 0.05 and absolute fold change ≥2 were considered differentially expressed genes (DEGs). All DEGs were mapped to GO terms and KEGG pathways in the respective databases. Significantly enriched GO terms and KEGG pathways were identified using hypergeometric tests and a Bonferroni correction with *p* < 0.05 as a threshold [77].

### 5.6. qRT−PCR Analysis

The transcript levels of genes were identified using a real−time quantitative polymerase chain reaction (qRT-PCR). Isolated micropores were extracted using TRIzol reagent (Invitrogen, GeneBetter Biotech, Beijing, China) following the manufacturer’s protocol. A Revert Aid First Strand cDNA Synthesis Kit (Vazyme Biotech Co, Nanjing, China) was used to reverse transcribe RNA to cDNA. The qRT-PCR reactions were conducted as reported previously, and *actin* (GenBank accession number XM_013731369.1; qRT-PCR forward primer: 5′-CCAGAGGTCTTGTTCCAGCCATC-3′, reverse primer: 5′-GTTCCACCACTGAGCACAATGTTAC-3′) was used as the internal reference gene in cabbage; the test was repeated three times [78]. We calculated the relative expression levels of the genes using the 2^−ΔΔCt^ method [79]. Significance was determined using a two-tailed Student’s *T* test (*p* < 0.05). The primer sequences of 9 genes were shown in Appendix A.

### 5.7. Association of miRNA, Transcriptome, and Methylation Sequencing Data

These association analyses in miRNA, transcriptome, and methylation were based on small RNA-Seq, RNA-Seq and MeDIP-Seq sequencing data. We filtered the raw datas to reduce the influence of sequencing error as described previously of the standard analysis. Firstly, the methylation regions of miRNA and mRNA were sorted out, and the methylation levels of miRNA and mRNA were annotated respectively. Then, we correlated miRNA target genes and differentially methylated regions (1000 bp slide windows, 500 bp overlap, FDR adjusted *p* < 0.05) with mRNA data (|log2FC| >1, FDR adjusted *p* < 0.05). We screened and sorted the data and regulatory relationships with significant differences among the three groups. Lastly, we performed GO and KEGG enrichment analyses of the associated core gene sets (hypergeometric tests and a bonferroni correction with *p* < 0.05 as a threshold).

## Figures and Tables

**Figure 1 ijms-23-05147-f001:**
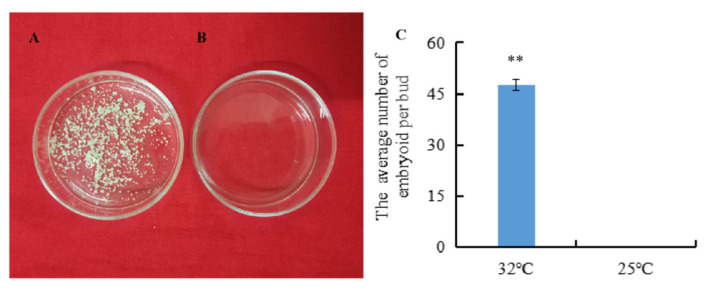
Embryoids produced by isolated microspores under the 32 °C and 25 °C pre-treatments. (**A**) Embryoids produced by isolated microspores cultured for 3 weeks after 24 h treatment at 32 °C; (**B**) Embryoids produced by isolated microspores cultured for 3 weeks after 24 h treatment at 25 °C; (**C**) The embryoid rates after the 32 °C and 25 °C treatments. Error bars represent standard error of the mean. Significant differences has been marked with asterisks (** *p* < 0.01).

**Figure 2 ijms-23-05147-f002:**
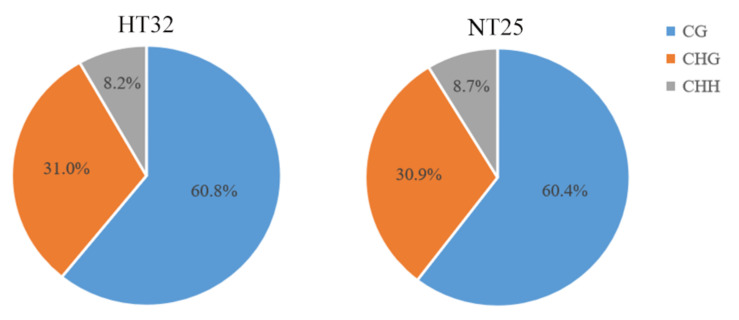
The cytosine methylation level (mC) percentage for each sequence context in HT32 and NT25.

**Figure 3 ijms-23-05147-f003:**
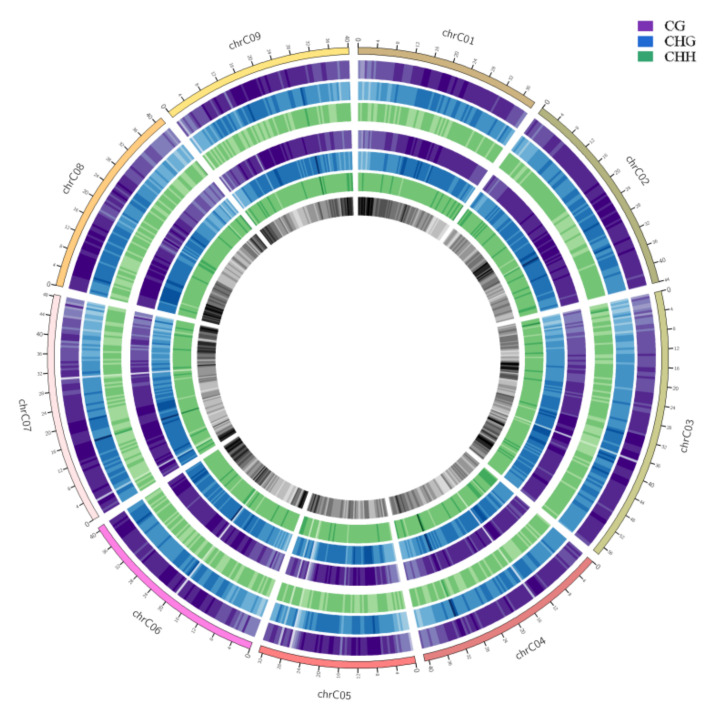
Genomic methylation levels of HT32 and NT25. The first circle represents chromosome name and scale. The second to fourth circles (purple, blue and green) represent the methylation background display of CG, CHG and CHH in the corresponding chromosome intervals of HT32 groups, respectively; The fifth to seventh circles (purple, blue and green) represents the methylation background display of CG, CHG and CHH in the corresponding chromosome intervals of NT25 group, respectively. The eighth circle represents the number of genes in the corresponding interval, and the darker the color, the more genes there are in that region.

**Figure 4 ijms-23-05147-f004:**
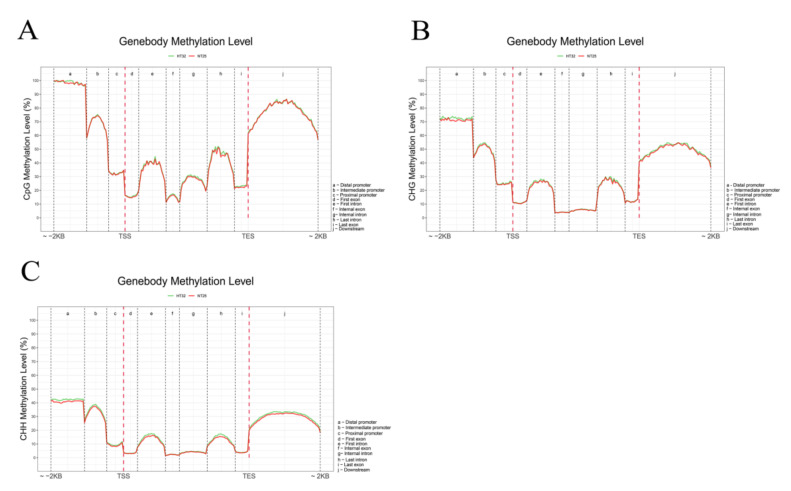
Comparative analysis of DNA methylation levels in different genomic regions between HT32 and NT25. (**A**) CG methylation level. (**B**) CHG methylation level. (**C**) CHH methylation level. Peak distribution in different genomic regions. Methylation density is the ratio of methylation peaks to the corresponding sequence lengths.

**Figure 5 ijms-23-05147-f005:**
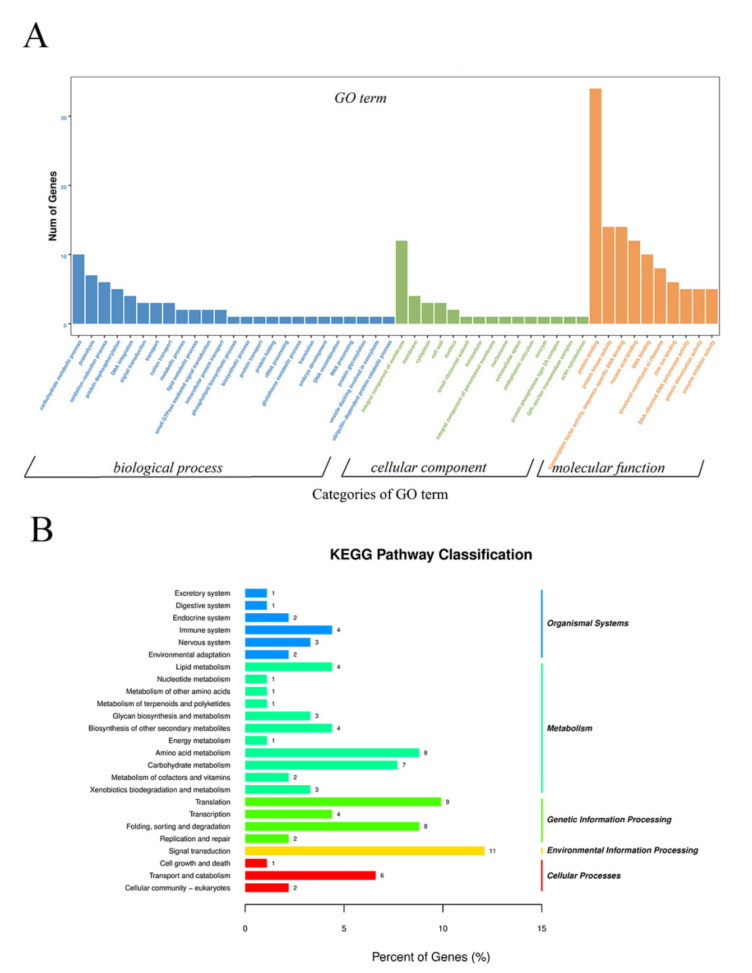
GO and KEGG enrichment of DMR-associated genes. (**A**) GO enrichment of DMR-associated genes. (**B**) KEGG enrichment of DMR-associated genes.

**Figure 6 ijms-23-05147-f006:**
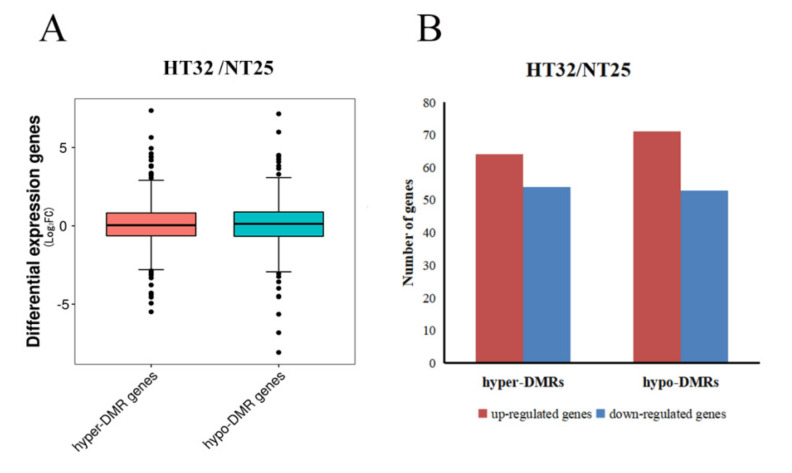
Correlation between differential methylation and level of transcript abundance. (**A**) Differential expression levels of genes associated with hyper/hypo DMRs. (**B**) The number of up/down-regulated genes associated with hyper/hypo DMRs.

**Figure 7 ijms-23-05147-f007:**
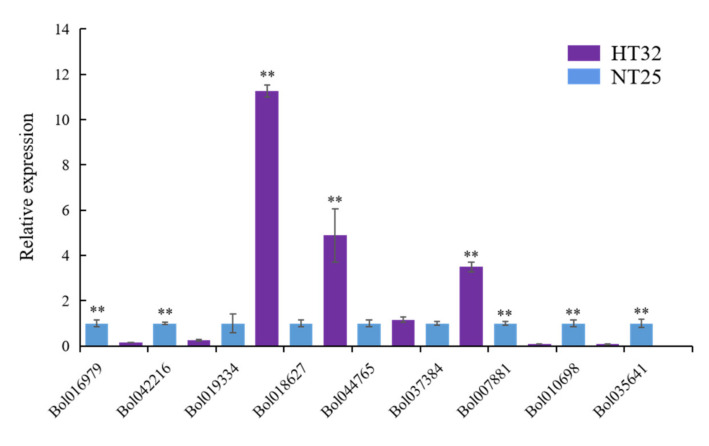
The qRT-PCR validation and expression analysis of 9 genes. The gene expression was normalized using *actin* and expressed relative to gene expression in HT32 sample (negative control). Error bars represent standard error of the mean. Significant differences has been marked with asterisks (** *p* < 0.01).

**Figure 8 ijms-23-05147-f008:**
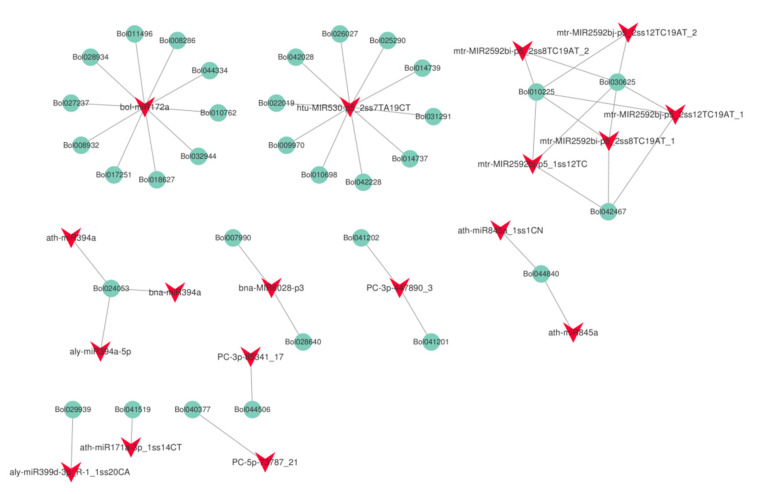
The interaction network of miRNA and target genes in plant hormone signal transduction pathway (ko04075).

**Figure 9 ijms-23-05147-f009:**
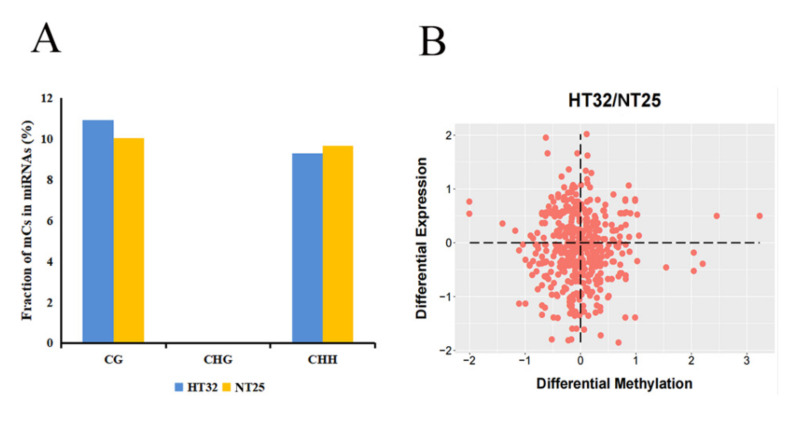
Correlation between differential methylation and level of miRNA abundance. (**A**) Fraction of different mCs present in miRNAs. (**B**) Differential expression levels of miRNAs associated with hyper/hypo DMRs.

**Figure 10 ijms-23-05147-f010:**
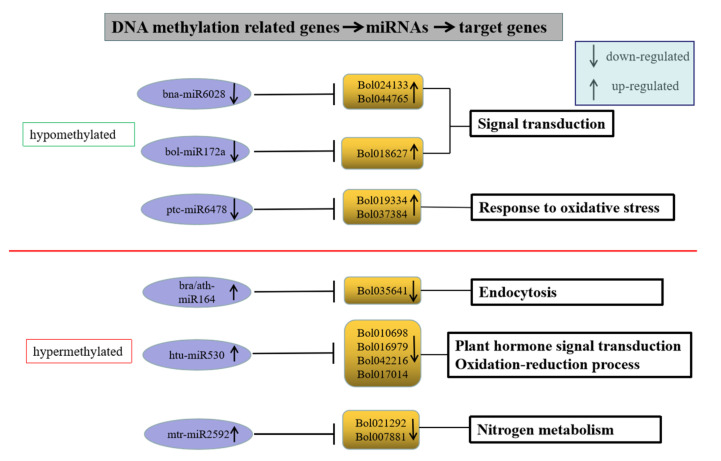
Genes with negative correlation with miRNA and methylome.

**Table 1 ijms-23-05147-t001:** Seven DEGs targeted by methylation and miRNA.

Target Genes	mRNA Regulation	Methylation Regulation	Location of DMR	miRNA Name	miRNA Regulation	*p* Value	Description
Bol007667	up	hypermethylation	promoter	PC-5p-73787_21	down	<0.05	plant U-box 9
Bol012681	up		downstream	bna-MIR6028-p3	down	<0.05	calmodulin-binding protein
Bol017014	down		downstream	htu-MIR530-p5_2ss7TA19CT	up	<0.05	atypical CYS HIS rich thioredoxin 5
Bol016003	up	hypomethylation	downstream	PC-5p-70200_22	up	<0.05	glycine-rich protein 2B
Bol045794	up		downstream	htu-MIR530-p5_2ss7TA19CT	up	<0.05	cyclic nucleotide-gated channel 6
Bol037384	up		downstream	ptc-miR6478_R-2	down	<0.05	peroxidase CB
Bol019334	up		downstream	ptc-miR6478_R-2	down	<0.05	peroxidase superfamily protein

## Data Availability

The raw reads of our BS-Seq, RNA-seq and miRNA-seq data in this work were deposited in the Sequence Read Archive under accession numbers PRJNA810282.

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
