# Peer review of "Global DNA Methylation and mRNA-miRNA Variations Activated by Heat Shock Boost Early Microspore Embryogenesis in Cabbage (Brassica oleracea)"

_ijms, 2022, doi:10.3390/ijms23095147_

Round 1

Reviewer 1 Report

The authors of the paper “Global DNA Methylation and mRNA−miRNA Profiles Reveal Key Events Activated by Short−term Heat Shock Boost Early Microspore Embryogenesis in Cabbage (Brassica oleracea)” analyzed the methylation levels of the genome, the differentially expressed genes, the miRNAs of microspores of cabbage after temperature treatments at 32°C (named as HT32) and 25°C (named as RT25) per 24 hours. Their study highlighted 1038 differentially methylated genomic regions (DMRs) in HT32. Furthermore, a total of 7,226 DEG (4,312 upregulated and 2,914 downregulated genes) transcripts were identified in both samples. The sequencing of miRNAs led to the discovery of 870 miRNAs in cabbage. Interestingly, the bioinformatic analysis showed the 31 DEGs were targeted by both methylation and miRNA. These changes may be correlated with the embryogenesis induced by heat shock treatment. Unfortunately, the paper is not ready to be published despite the good results. The material and methods section has to be improved as well as the quality of the figures. The results are to be shown in a better way. The authors are invited to resubmit a revised version of the paper.

Here are some tips and questions of mine:

  1. Recheck the grammar of the title. It might be better to use the gerund “boosting” instead of boost.
  2. In the abstract section (at line 19) you wrote “Global methylation levels were significantly different in the two treatments” but your data seem to suggest that there is no difference.
  3. At line 59, what did you mean for decreases in DNA methylation 1? A reduction of its activity?
  4. In line 1, use the italics font for Brassica oleracea capitata.
  5. Indicate a reference for the statements at lines 37-39.
  6. In line 43, use italics for in vitro.
  7. Italic font at lines 46 (Brassica napus) and 47 (B. oleracea).
  8. Maybe, there is a typo at line 77 (gibberellic acid MYB family genes). What did you mean for gibberellic acid in reference to MYB family genes?
  9. Why did you focus on miR159 and miR167 in the introduction? Are they quite commonly studied in miRNA profiling?
  10. Improve the caption of figure 1.
  11. You might highlight that the microspores, shown in figure 1, were obtained after three weeks from the two temperatures treatments. So, you could add this information in the caption of figure 1.
  12. Specify how you performed the statistical analysis for figure 1C. Did you use the t-student test?
  13. Remove the indication “24h” from the x-axis of figure 1C.
  14. There is a typo in the label of the y-axis of figure 1C. Embryos/ bud instead of Embry/bud
  15. Did you sequence the genome of the microspores which were obtained with the standard protocol and then they were subjected to 25°C or to 32°C for 24 hours?
  16. Describe better what you mean for the acronyms HT32 and RT25, at line 108.
  17. It is not clear to me which is the difference between the analysis shown in figures 2A and 2B.
  18. At lines 113 and 116, mCs are equal to 32351.452 and 42291.691 in HT32 and in CT25, respectively. After at lines 116-117, you reported that the total mCs were identical. So could you explain this better?
  19. How many biological replicates were sequenced in BS-seq experiment?
  20. Indicate the link of the reference genome at line 112.
  21. Are the differences between the levels of CG, CHG, and CHH methylation, in the whole cabbage genome (figure 2B), statistically significant?
  22. Indicate in the caption of figure 2 the level of C that is shown in figure 2B.
  23. Show the standard deviation in figure 2B.
  24. At line 135, could you indicate the chromosomes with a different level for mCHH between HT32 and RT25? You should describe better the results shown in the circle diagrams.
  25. Improve figure 3. Maybe you can show only the results of figure 3A improving its quality. You might show the legend of figure 3A. Instead, you might indicate figure 3B in the supplementary section, and you could show the results for the three types of methylation of RT25 an HT32 separately.
  26. Are 1038 differentially methylated genomic regions (DMRs) found only in HT32 samples?
  27. Improve the quality of figure 5. It is not readable the x-axis of figure 5A.
  28. Did you perform the GO and KEGG analysis for the RT25 samples?
  29. Could you show the 7,226 DEGs of HR32 and RT25 in the supplementary section?
  30. Could you show a Ven diagram of DEGs for HT32 and RT25? The same analysis for up and down DEGs?
  31. In line 302, there is a typo “HR32”.
  32. There is a typo in figure 6A. HT32 instead of HT3R. Furthermore, there is an extra dot in the names indicated on the x-axis.
  33. Are you showing the log2FC of DEG on the y-axis of figure 6A?
  34. Could you show the 63 differentially expressed miRNAs 219 (DERs) in the supplementary files?
  35. In line 232, what did you mean for “ko04075”? Is the GO term for the signal transduction pathway that you selected?
  36. In line 251, why did you write that the genes are negatively affected if they are up-regulated when hypomethylated?
  37. Maybe the expression “pairs of genes” is not properly right.
  38. The figure 9, there is a typo. It misses a space between the words signal and transduction.
  39. Improve the description of the caption of figure 9.
  40. In line 269, you stated that seven genes out of 31 DEGs were found to be closely related to early microspore embryogenesis. Which criteria did you use for this statement?
  41. Typo at line 290, 32HT.
  42. In the caption of figure 10, indicate the reference gene (actin) used for the real-time study. Is actin known to be stable during heat stress?
  43. Furthermore, indicate the calibrator samples whose relative expressions are equal to 1 in the caption of figure 10. I imagine that you set the RT25 as calibrator samples for every tested gene.
  44. You have to include a statistical analysis for the qRT-PCR analysis.
  45. There is a typo in Table 1. Target instead of “taget”.
  46. Could you indicate the 63 differentially expressed miRNA highlighting the 18 one which has a threefold relative change? You might show it in the supplementary table.
  47. At lines 307 and 308, are the difference in the genomic methylation level in mCHH and in mCG, slightly lower and higher in HT32, statistically different?
  48. Typo at line 312, use italic font for Brassica napus. Check all the text.
  49. Could write a short conclusion after the discussion.
  50. In line 363, you might improve the title of paragraph 4.1. Treatments or heat shock treatment instead of treatment.
  51. At line 367, indicate the full name of the abbreviation BS-Seq.
  52. Cite a reference for the NLN−13 medium at line 369. Report its chemical composition.
  53. It is not clear to me the methods that you used for obtaining the microspores. Did you use the standard protocol for Brassica oleracea microspore culture using a heat shock pretreatment and NLN-13 medium (Lichter 1981)? After that, were the obtained microspores subjected to two different temperature treatments for 24 hours?
  54. Specify better what you mean for HT32 (32°C, 24 h microspores) and RT25 (25°C, 24 h microspores) in paragraph 4.1.
  55. You do not describe how you cultivated the embryoids (embryos per bud) for three weeks after the two temperature treatments.
  56. In line 373, change the term “part”.
  57. The sentence at line 381 is not completed.
  58. It seems to me that you did not describe the BS-seq methods in paragraph 4.2.
  59. Describe better the six small RNA libraries (HT−1,2,3 and RT−1,2,3) in paragraph 4.3.
  60. At line 462, which threshold did you use for GO and KEGG analysis?
  61. Report the unigene number for the actin in the material and methods section.
  62. Did you test the efficiency of the primer pairs?
  63. Indicate in the material and methods that the sequence of the primer pairs is indicated in the supplementary section.
  64. Indicate in the material and methods how you performed the statistical analysis s for all your results.
  65. Report how you performed the network analysis in the material and methods. I can not find its description.
  66. Describe the methods of the circle graph in the material and methods.

Reviewer 2 Report

Your manuscript entitled “Global DNA Methylation and mRNA−miRNA Profiles Reveal Key Events Activated by Short−term Heat Shock Boost Early Microspore Embryogenesis in Cabbage (Brassica oleracea)” has focused on the understanding of the relationship among global DNA methylation and small RNA and gene expression in heat shock induced microspore embryogenesis of cabbage. Your manuscript contains more interesting information in this research topic which can support the practical breeding programmes in some species of cruciferous family.

In the Introduction, the authors have summarised more important published information in connection with their research topic. However, the cabbage belongs to the cruciferous family, as they mentioned. In this family, rapeseed is the most important species, which is the standard species in the study of microspore embryogenesis, especially “Topas” genotype. In the Introduction of the manuscript, they have to summarize the most important published data in this topic. The authors should detail the most important results of their earlier publication, which has focused on similar research topic in rapeseed than this publication in cabbage, and they can detail other publications from other researchers.

Li, J.; Huang, Q.; Sun, M.; Zhang, T.; Li, H., Chen, B.; Xu, K.; Gao, G.; Li, F.; Yan, G.; Qiao, J.; Cai, Y.; Wu, X. Global DNA 533 methylation variations after short−term heat shock treatment in cultured microspores of Brassica napus cv. Topas. Sci. Rep. 2016, 534 6, 38401. DOI: 10.1038/srep38401.

Page 1 line 43: Pretreatment is written in two different ways in the manuscript (pre-treatment, pretreatment). You should write pre-treatment instead of pretreatment.  

Figure 1: The number of embryos is mentioned in the text of the manuscript, while the rate of embryiod can be read in the diagram (Figure 1). You have to use same correct expression for the microspore – derived structures (embryo or embryoid).    

Results:

Page 11 line 274-286: This part belongs to the Discussion in this form. You have to reconsider this part of your manuscript.

Discussion:

In the Discussion (Page 13 line 312), you have to compare in more detail the data of your experiment in cabbage with the earlier published data of similar experiments in rapeseed, for example your publication (Li et al. 2012) etc.

Page 12 line 303: You should write microspore instead of pollen.

Your manuscript contains interesting and useful information in connection with microspore embryogenesis. After some improvement of manuscript, the decision of your manuscript can be reconsidered.

Reviewer 3 Report

The article with the title "Global DNA Methylation and mRNA-miRNA Profiles Reveal Key Events Activated by Short-term Heat Shock Boost Early Microspore Embryogenesis in Cabbage" is handling microspore breeding improvement. THe research focus is on the reason of heat shock mechanisms improving the early microspore embryogenesis. The results showed 508 differentially methylated regions (DMRs) and also 31 differentially expressed genes (DEGs) affected by miRNA or methylation changes. In the functional characterization of DEGs the publication showed some ROS and ABA involved genes.

Abstract:

  1. Unclear are the heat shock conditions (short in the title) but no temeprature or time is described in the abstract.
  2. Missing is the function of the remaining 27 genes as only 4 are explained. Because there are no further experiments and based on the descriptive character of the article this has to be mentioned in the abstract or not put so much attention.

Introduction:

  1. The section about heat shock in microspore embryogenesis is not very clear and has to be prolonged and described. Missing is the normal temperature as well as a clear differentiation between specific species and timing of heat shock. At stage the sentences are not very easy to understand.
  2. The paragraph about the methylation is not really good connected to heat stress. First DNA methylation is connected to short stresses or fast dynamic processes but the stress of the authors was 1-2 days based on the first paragraph. Also the connection to heat stress in cotton and the one example is not very informative and there should be more examples and clearlity in the onnection from methylation to heat shock.
  3. The general heat stress response based on HSPs and HSFs is not mentioned at all and also splicing as an additional source of HSR is not mentioned here. So it should be more clearly emphasized what is expected to analyze and combine the two part miRNA and methylation.
  4. The last paragraph should explain more the experimental design and the aims of the study. So why is miRNA and methylation analyzed and how to connect both? What is happening with the mentioned 32.5°C? What is exactly analyzed in how many replicates? How long was the stress (stress regime information).

Results:

  1. using two temperatures resulting in one (25°C) with zero embryos per bud at all is hard for comparison. It is not clear why not different temperatures have been used between 25 and 32 degree or even higher to get a clearer picture. THis has to be extended for some results.
  2. How can be microspore NGS analysis without any microspore (Figure 1B)? There is no material for the analysis named RT25.
  3. With only 1 replicate for the methylation it is hard to differntiate between confounder effects and real effects based on heat shock. Biological replicates as well as technical replicates are missing.
  4. More prolbematic is the use of a cultivar (01-88) and mapped against a refernce 02-12. This problem can lead to false negatives if methylation sites are not present or false positive by mutations in the genomes which are incorrectly interpreted as BS-seq changes from T to C.
  5. From the numbers from figure 2 and the not readable Figure 3 it is hard to see any differentially methylated sites.
  6. The small changes (less then 1-5%) between the overall chromosomes for RT25 and HT32 can be not called differntially methylated without a statistical testing. This overall small changes could be also due to higher or lower sequencing depth which was in the results stated in the beginning and the normalisation procedure was not explained.
  7. The wording about differential methylation pattern of gene bodies is misleading as it is at stage not verified or validated that there is a differntial methylation. THe changes are marginal and without testing there is not such statement possible.
  8. The scaling of the gene bodies is not clear. As for example the length of last exons varies drastically it has to be explained how now the category I (X-axis) is used or has to be read. If it is a single region of a specific gene it would be fine but the separation of I to J and the length and positioning (Scaling to 100%?) has to be explained and more analyzed in detail.
  9. BAsed on a single replicate what was done to find differntial methylation sites. A in silio approach or simulated datasets have to be added for standard tests to have at least two replicates but even here it has to be shown that this is working in this scenario. THe authors need more experiments or have to weaken their statements to hypothesis.
  10. The enrichment analysis for functional annotation is not very detailed explained based on the different subcategories. Here the functional part and some more information in the text are needed.
  11. The DEGs have to be explained. The state of the art is the usage of DESeq2 or EDGER and both perform a shrinkage and need at least two replicates per condition. Both is not used here as the p-value includes because of the multiple testing error a lot of false positives up to 1000 already by their 10.000 tests. The authors have to use the adjusted p-value and the log2 FC cannot be used anymore as the shrinkage is already changing the scaling and this is also already intrinsically used in the DESeq2 and EdgeR analysis.
  12. How was the DMR connceted to the DEGs? It has to be clarified if intronic and other parts are also included for DMRs as they have not necessarily an influence on the DEG of a specific mRNA.
  13. The figure 6 is clearly showing, that the DMRs seem to have no effect on the DEGs if the A part of the figure is meaning the foldchange on the Y-axis. There is no difference for them in the fold change except of outliers. The authors have to show the distribution as well for the not up and down regulated genes.
  14. For the miRNAs there were replicates perofrmed and named in the text. So for them a differntial miRNA expression could be performed. Here it is important to explain how the 870 miRNAs have been identified. If they are the annoteated ones from the genome source it would be fine. In the other cases based on predictions it has to be mentioned about the reference database and algorithm.
  15. The target identification and what targets for the miRNA is not described (methylation positions or mRNA expression)
  16. The miRNAs are also named after other species like A. thaliana (ath) meaning there was a mapping included leading to putative miRNAs but not necessarily real miRNAs.
  17. Also the bar charts for the mCs in miRNAs show no significant changes (plots with no p-values). So overall the results are not convingly showing any differences between the two treatments.
  18. The last paragraph about validation of gene expression should belong in the first paragraph about the gene expression in general and is a control for the supplements but not essential in the main text before the discussion.
  19. The finding of 31 DEGs but only describing 4 in the abstract and naming 7-8 in the table of the results is hard to understand. Also the different variables are not clearly describing an effect. Ther are hypermethylated genes names with mRNA upregulation and miRNA downregulation in different regions (promoter and downstream). How is this now related to a real effect in the microspore embryogenesis?
  20. The remaining genes are not described or analyzed or discussed a t all.

Discussion:

  1. Overall the discussion is not related to the state of the art for heat stress or heat stress response. Known key regualtors are not even vcontrolled or set in the context like HSPs or HSFs.
  2. Most of the genes identified in this study are simply ignored without giving reasons for this.
  3. The general identity between the different temperatures show already that there is only a marginal effect which is not correlating to the drastic effect in the phenotype of figure 1.

Methods:

  1. In the methods the authros write something about replicates but the results are not giving any details about the numbers from the single replicates and the correltation between the same treatments.
  2. Also the heterogeneity or homogeneity between DNA and RNA of the different points of time (stating that the experiment was performed three times and both was harvested simultaneosly) is not mentioned but has to be controlled to correct for batch effects and confounders.
  3. Differences of the replicates for the methylation were not analyzed or mentioned. So how many differnces have been observed already between the 25° replicates?
  4. For miRNA a in-house program was used but it is not refernced but a refernce to a ACGT101-miR is given? THe appraoch for the processing of miRNAs is not based on standard software or known and published appraoches and by that has to be explained in much more detail or replaced by a standard approach teh authors can reference.
  5. For differential miRNA or mRNA no standard package like DESeq2 or EdgeR was used. This lead to a very high Flase positive and False negative reate and has to be changed. See comments above.
  6. Also for the GSEA the tool and procedure and statistcal process is missing.
  7. The authors have been setting own threshold to call something correlated or significant. In the association of all of them miRNA, mRNA and methylation this is not possible without clear testing and network creation. Also the target identification is missing an dregion categorization for some of the figures.

Round 2

Reviewer 1 Report

The authors performed an integrated genome-wide analysis of DNA methylation, miRNAs, and mRNA transcriptional activity, using heat-stressed (32°C for 24 h) and control (25°C for 24 h) microspores from cabbage. They answered all the questions comprehensively and they improved the manuscript. The introduction provides a better background in the new version of the manuscript, as well as the results and material and methods section show a better presentation. Despite their efforts and the results obtained, the manuscript needs to be further improved. The results should be shown in a clearer way in some points of the text.

Here are some tips of mine:

The title is not properly right. It should be better not to highlight the discovery of key events.

You might change the expression “non-HS stress” in control pre-treatment. It is up to you.

It might be better to use the term pre-treatment instead of treatment at lines 133 and 134.

The same above consideration for the caption of figure 1, you might use the term pre-treatment instead of treatment.

Sorry, it is still not clear the difference between the analysis shown in figures 2A and 2B. It seems to me that you showed the same results in two different ways (your answer: “Figures 2A shows the proportion of three types of cytosine methylation sites. Figure 2B shows the proportion of cytosine methylation sites in the cytosine sites”). You might remove figure 2B and show it in the supplementary section.

Recheck the sentence at lines 198-200, please.

In line 222, you should not use the term genes. You might use the term DMR-associated genes. This is up to you.

The paragraph “2.4. Validation of gene expression” should be postponed after paragraph 2.5.

You should improve the caption of figure 6. You should indicate the housekeeping gene that you used as a reference. It should be clear which is your calibrator whose relative expression is equal to 1.

Paragraph 2.5 is not clear. Postpone the sentence at lines 252-253 after the description of the results of RNA seq.

Recheck the sentence at lines 297-299.

Rewrite the sentence at lines 337-340.
